# Extensive allele-specific translational regulation in hybrid mice

Jingyi Hou[1,†], Xi Wang[1,†], Erik McShane[2], Henrik Zauber[2], Wei Sun[1], Matthias Selbach[2] & Wei Chen[1,*]

## Abstract

Translational regulation is mediated through the interaction between diffusible *trans*-factors and *cis*-elements residing within mRNA transcripts. In contrast to extensively studied transcriptional regulation, *cis*-regulation on translation remains underexplored. Using deep sequencing-based transcriptome and polysome profiling, we globally profiled allele-specific translational efficiency for the first time in an F1 hybrid mouse. Out of 7,156 genes with reliable quantification of both alleles, we found 1,008 (14.1%) exhibiting significant allelic divergence in translational efficiency. Systematic analysis of sequence features of the genes with biased allelic translation revealed that local RNA secondary structure surrounding the start codon and proximal out-of-frame upstream AUGs could affect translational efficiency. Finally, we observed that the *cis*-effect was quantitatively comparable between transcriptional and translational regulation. Such effects in the two regulatory processes were more frequently compensatory, suggesting that the regulation at the two levels could be coordinated in maintaining robustness of protein expression.

**Keywords** allele-specific gene expression; *cis*-elements; translational regulation

**Subject Categories** Genome-Scale & Integrative Biology; Protein Biosynthesis & Quality Control

**Mol Syst Biol. (2015) 11: 825**

## Introduction

Eukaryotic gene expression is orchestrated by multiple regulatory processes, of which one critical step is mRNA translation. While mRNA abundance levels are widely used as a proxy of protein expression, yet, in various eukaryotes, only up to 50% of variation in protein level can be explained by that in mRNA abundance (De Sousa Abreu *et al*, 2009). Recent genome-wide studies further highlight the predominant role of translation in controlling cellular protein concentrations, in both yeast and mammalian cells (Schwanhäusser *et al*, 2011; Marguerat *et al*, 2012). Translational regulation, accounting for not only rapid response during stress but also long-term adaptation in cell physiology (Sonenberg & Hinnebusch, 2009; Spriggs *et al*, 2010), is mediated via the interaction between the *cis*-regulatory elements residing in the mRNA transcripts and various *trans*-factors (e.g. translational machinery, RNA binding proteins (RBPs) and miRNAs). Previous studies have reported a variety of *cis*-elements involved in translational regulation, including Kozak sequence (Kozak, 1986), upstream open reading frames (uORFs) or upstream AUG codons (uAUGs) (Mueller & Hinnebusch, 1986; Matsui *et al*, 2007; Calvo, 2009), and binding sites of miRNAs and different RBPs (Hentze *et al*, 1987; Leibold & Munro, 1987; Abaza & Gebauer, 2008; Fabian, 2010). Genetic variants disrupting these *cis*-elements often alter protein synthesis and result in pathological phenotype (Cazzola & Skoda, 2000; Signori *et al*, 2001; Beffagna *et al*, 2005).

Changes in translational regulation represent one of the major dynamic processes during evolution, and such changes could arise from the divergence in *cis*-regulatory elements. Compared to transcriptional regulation, where numerous genome-wide studies have been using first microarray and then deep sequencing to dissect *cis*-regulatory divergence in different organisms, global analysis of translational *cis*-regulation is rather limited. Recently, similar to expression quantitative trait locus (eQTL) mapping in the study of transcriptional regulation, genome-wide mapping of protein quantitative trait loci (pQTLs) has been performed to investigate genetic variants responsible for inter-individual variation in protein abundance (Ghazalpour *et al*, 2011; Skelly *et al*, 2013; Wu *et al*, 2013; Battle *et al*, 2015). For instance, using mass spectrometry (MS)-based shotgun proteomics approach, Ghazalpour *et al* (2011) quantified over 5,000 peptides in 97 inbred and recombinant mouse strains and identified 46 local pQTLs for 396 genes. Using an improved MS-based approach, Wu *et al* (2013) determined relative protein levels for 5,953 genes in human lymphoblastoid cell lines (LCLs) from 95 individuals and identified 77 genes with local pQTLs. In both studies, despite the overlap between some pQTLs and eQTLs, approximately half of the pQTLs cannot be explained by mRNA expression divergence (Ghazalpour *et al*, 2011; Wu *et al*, 2013). This suggests that genetic variants contribute substantially to inter-individual difference in protein abundance only by affecting post-transcriptional processes. Very recently, taking advantage of ribosome footprinting technique (Ingolia *et al*, 2009), in addition to

1   Laboratory for Functional and Medical Genomics, Berlin Institute for Medical Systems Biology, Berlin, Germany
2   Laboratory for Proteome Dynamics, Max-Delbrück-Centrum für Molekulare Medizin, Berlin, Germany
    *Corresponding author. Tel: +49 30 94062995; E-mail: wei.chen@mdc-berlin.de
    †These authors contributed equally to this work

eQTL and pQTL profiling, Battle *et al* (2015) mapped the genetic variants that are associated with individual specific difference in ribosome occupancy (rQTL) to more directly dissect the impact of genetic variants on translation. Based on their data obtained from 72 human LCLs, among 4,000 genes quantified for all three phenotypes, 90 and 35% of rQTLs and pQTLs overlapped with eQTLs, respectively (Battle *et al*, 2015).

An alternative approach that could more directly address the *cis*-effect is to compare the allelic difference in an F1 hybrid, where mRNA transcripts from both parental alleles are subject to the same *trans*-regulatory environment; thus, observed allele-specific pattern should only reflect the impact of *cis*-regulatory divergence. Recently, based on ribosome footprinting technique, this approach has been used to investigate allele-specific translational efficiency (TE) in F1 hybrid yeast (Albert *et al*, 2014; Artieri & Fraser, 2014b; McManus *et al*, 2014). While all these studies revealed a pervasive *cis*-regulation at the translational level, which is comparable to the *cis*-effect at transcription, it is controversial whether allelic translational regulation more frequently compensates or reinforces the divergence resulting from allele-specific transcription. Compared to unicellular organisms, more complex regulation is required in multicellular species. However, genome-wide profiling of allele-specific translational pattern in any of them is still lacking.

In this study, to globally investigate *cis*-divergence in translational regulation in mammals, we applied mRNA sequencing and deep sequencing-based polysome profiling to quantify the allele-specific TE in an F1 hybrid between two inbred mouse strains, *Mus musculus* C57BL/6J (B6) and *Mus spretus* SPRET/EiJ (SPRET). The two parental strains chosen in this study diverged ~1.5 million years ago, which results in ~35.4 million single nucleotide polymorphisms (SNPs) and ~4.5 million insertion and deletions (indels) between their genomes (Keane *et al*, 2011). Such a high sequence divergence allowed us to unambiguously determine the allelic origin for a large fraction of sequencing reads, thereby enabled accurate quantification of allelic TE for thousands of genes. Out of 7,156 genes with reliable quantification of both alleles, we identified 1,008 genes (14.1%) with significant allelic biases in TE. Compared to genes without allelic bias, those with bias in TE contained higher density of sequence variants, particularly in the 5′UTR regions, including those affecting local RNA secondary structure in vicinity of start codon or changing proximal out-of-frame uAUGs. Finally, we observed quantitatively comparable allelic divergence in transcription and translation. Consistent with previous reports that the abundance of protein tends to be less diverged than that of RNA across different species, allelic biases in the two processes were more frequently compensatory.

## Results

### Pervasive allelic divergence in translational efficiency (ADTE)

To investigate the allelic divergence at the translational level in a mammalian system, we derived fibroblast cell lines from an F1 hybrid mouse between C57BL/6J and SPRET/EiJ strains. Using the F1 fibroblasts, we deep-sequenced the polyadenylated RNAs to measure mRNA abundance (total mRNA) and, in parallel, performed deep sequencing-based polysome profiling to estimate the translational status by quantifying the abundance of mRNA transcripts associated with polyribosome (poly-mRNA) (Fig 1A; see Materials and Methods for details). From two biological replicates, paired-end sequencing of total mRNA and poly-mRNA produced on average 158.5 and 94.6 million 100-nt read pairs, respectively (Table EV1 and Fig EV1). The high density of sequence variants between the genomes of C57BL/6J and SPRET/EiJ allowed unambiguous assignment of allelic origin for an average of 61% total mRNA and 65% poly-mRNA uniquely mapped reads (Table EV1 and Fig 1B; see Materials and Methods for allelic read mapping).

We defined translational efficiency (TE) as the abundance ratio between poly-mRNA and total mRNA, and used only the reads assigned with unambiguous allelic origin to assess the allele-specific TE in a quantitative manner. More specifically, we used only the reads that were mapped on the SNP loci in protein-coding regions. After filtering out the SNP loci with potential allelic read mapping biases due to incomplete SNP annotation in paralogous or pseudogenes (see Materials and Methods for details), 7,156 genes containing at least five coding SNPs supported with sufficient allelic reads were retained (see Materials and Methods for details). Figure 1C showed two representative examples with significant ADTE, biased towards C57BL/6J and SPRET/EiJ allele, respectively.

To further formally determine the genes with significant ADTE, while accounting for the non-uniform allelic read counts at different SNP loci across the same genes, we applied a bootstrapping strategy to estimate the confidence of calculated allelic TE ratio, as previously used by Muzzey *et al* (2014) (see Materials and Methods). In brief, for each gene consisting of a list of at least five coding SNPs, we generated 5,000 new lists, each comprised of the same number of SNPs that were chosen at random with replacement from the original list. For each of the 5,000 random list, allelic TE ratio was calculated and altogether yielded a bootstrap distribution, which was then summarized with a mean and a standard deviation. The larger the bootstrap mean deviates from zero, the larger the TE diverges between the two alleles. By contrast, lower bootstrap standard deviation gives more confidence in the estimation of allelic TE ratio. As shown in Fig 2A, 81 and 98% of all analysed genes showed a bootstrap standard deviation lower than 0.2 and 0.4, respectively, indicating the good quality of our total mRNA and poly-mRNA data. Based on the bootstrap mean and standard deviation, the statistical significance of ADTE was then determined for each gene (Fig 2A; see Materials and Methods). After applying a threshold of Benjamini–Hochberg-adjusted *P*-value < 0.05 and allelic TE divergence > 2.0 in both replicates (FDR = 4.85%, Fig EV2A), we identified 1,008 (14.1%) genes exhibiting significant ADTE.

To assess the accuracy in quantifying ADTE based on short Illumina reads, we randomly selected 33 genes for independent validation. Using the PacBio RS system, we deep-sequenced the RT–PCR products (500–600 bp, spanning ≥ 3 SNPs) amplified from both total mRNA and poly-mRNA using primers targeted at the regions with no sequence variant between the two alleles (see Materials and Methods) (Eid *et al*, 2009; Sun *et al*, 2013). The longer read length facilitated the assignment of the PacBio reads to the parental alleles without any ambiguity. Allelic ratios of both total mRNA and poly-mRNA abundances could therefore be calculated with high precision. As shown in Fig 2B, the ADTE estimated in this way was significantly correlated with that determined by Illumina approach ($R^2 = 0.912$, $P < 10^{-17}$).

**Figure 1.** **Deep sequencing-based global quantification of allele-specific translational efficiency.**

A   Study design. Fibroblast cell line was derived from an F1 hybrid mouse between C57BL/6J and SPRET/EiJ inbred strains. Using the F1 fibroblasts, we deep-sequenced the polyadenylated RNAs to measure mRNA abundance (total mRNA) and, in parallel, performed deep sequencing-based polysome profiling to estimate the translation status by quantifying the abundance of mRNA associated with polyribosome (poly-mRNA).

B   The percentage of uniquely mapped reads from total mRNA sequencing (left) or polysome profiling dataset (right) that were unambiguously assigned to C57BL/6J (red) and SPRET/EiJ (blue) alleles, or assigned to the two alleles with equal probability (common, grey). On average, 61% total mRNA and 65% poly-mRNA uniquely mapped reads could be unambiguously assigned to either allele. See Table EV1 for the detailed statistics of allelic read mapping.

C   Barplots showing the number of sequencing reads from total mRNA sequencing (mRNA) or polysome profiling dataset (Poly) assigned to C57BL/6J (red) or SPRET/EiJ (blue) alleles (y-axis) at different SNP loci (x-axis) across the coding region of genes *Cnppd1* (up) and *Lbp* (low). In *Cnppd1*, whereas the mRNA transcribed from the two alleles was of similar abundance, mRNA associated with polysome contained higher amount of C57BL/6J-derived transcripts, indicating the higher translational efficiency of C57BL/6J allele. In contrast, transcripts derived from the C57BL/6J allele of gene *Lbp* was translated at lower efficiency than SPRET/EiJ-derived transcripts.

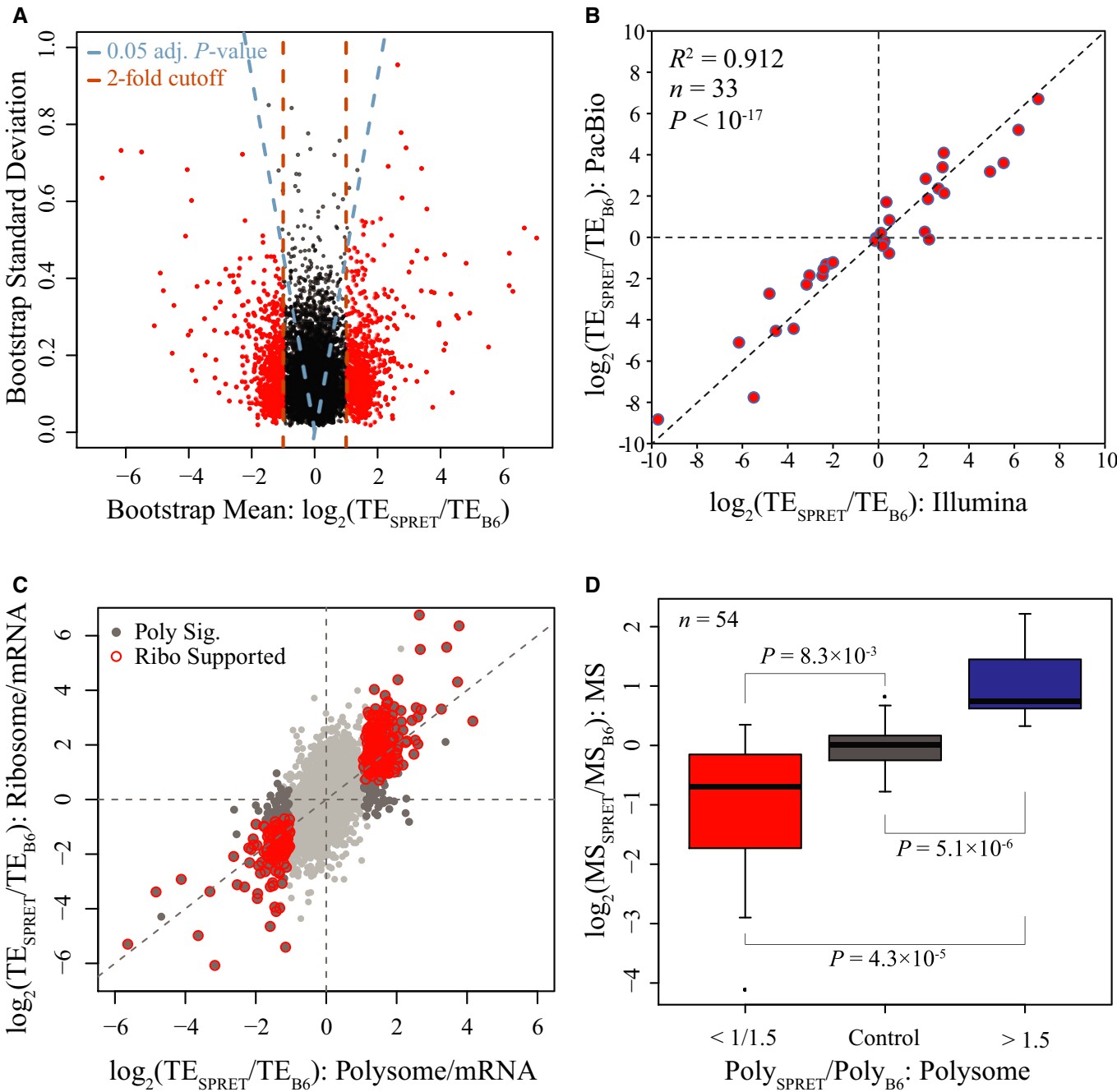

**Figure 2.   Identification of genes with significant ADTE.**

A   Scatterplot showing the bootstrap means (*x*-axis) and standard deviations (*y*-axis) in estimating ADTE for the 7,156 genes containing at least five coding SNPs supported with sufficient allelic reads. Dashed blue lines indicate the Benjamini–Hochberg-adjusted *P*-value of 0.05, and dashed brown lines indicate the twofold divergence. Genes with significant ADTE (Benjamini–Hochberg-adjusted *P*-value < 0.05, allelic TE bias > 2-fold) are depicted in red.

B   Scatterplot comparing ADTE estimated based on Illumina sequencing data (*x*-axis) to that based on PacBio sequencing (*y*-axis) for the 33 randomly selected genes. The ADTE estimated based on PacBio sequencing was significantly correlated with that determined by Illumina approach ($R^2 = 0.912$, $P < 10^{-17}$).

C   Scatterplot comparing the ADTE estimated based on polysome profiling data (*x*-axis) to that based on ribosome footprinting data (*y*-axis). All dots represent the 4,511 genes with both sufficient polysome profiling and ribosome footprinting data. Among them, the 688 genes with significant ADTE based on polysome profiling data are depicted in dark grey, of which the 460 genes that were also estimated with significant ADTE based on ribosome footprinting data are depicted in red circles.

D   Boxplots showing the distribution of allelic bias in protein abundance estimated using mass spectrometry (MS)-based proteomics approach. The 54 genes that were confidently quantified for their allelic protein abundance using MS approach were categorized into three groups according to polysome profiling data, that is no allelic bias (*n* = 35), bias towards C57BL/6J (*n* = 10) and SPRET/EiJ allele (*n* = 9). The allelic biases estimated using MS approach were on average coherent with that based on polysome profiling data, and that the MS estimates were significantly different among all the three groups (*P* < 0.05 for all pairwise comparisons, Mann–Whitney *U*-test).

As another independent approach, we also performed ribosome footprinting to assess mRNA translational status. In comparison with polysome profiling that measures the relative abundance of mRNA transcripts in the active translating pool, ribosome profiling directly measures the number of ribosomes associated with different mRNAs and therefore in principle enables more precise estimates of protein synthesis rate (Ingolia *et al*, 2009). The insert size of the ribosome profiling library was limited by the length of ribosome-protected mRNA fragments (RPFs), that is 28–33 nt. Therefore, the library was sequenced only for 50 nt. After trimming adapters, we mapped 165.9 million RPF reads to the reference sequences of both B6 and SPRET transcriptome in the same manner as we did with the total mRNA and poly-mRNA data. Due to the short length after adaptor trimming, only 19% uniquely mapped RPF reads could be unambiguously assigned to either allele (Fig EV2B and Table EV1). Consequently, only 4,511 ORFs consisting of $\geq 5$ SNPs supported with sufficient allelic RPF reads could be used for ADTE calculation. Applying the same bootstrapping strategy, we identified 1,305 genes with significant ADTE (Benjamini–Hochberg-adjusted *P*-value < 0.05; Fig EV2C). Among the 1,008 genes with significant ADTE identified based on polysome data, 688 had sufficient allelic ribosome profiling data. Among them, 460 genes (66.9%) showed also significant ADTE bias towards the same allele as estimated based on polysome profiling (Fig 2C; see also Fig EV3A for comparing the allelic divergence in translational status estimated based on polysome profiling data versus that based on ribosome footprinting data). Importantly, no single gene showed significant ADTE but towards the different allele between polysome profiling and ribosome footprinting results.

We also used mass spectrometry (MS)-based proteomics to directly quantify protein abundance. To minimize the influence of protein degradation, we measured only newly synthesized proteins using azidohomoalanine (AHA) labelling, which in principle provides a more direct proxy for translational status than polysome or ribosome profiling (Dieterich *et al*, 2006). Due to much lower number of peptides that could be detected and assigned to either allele, 54 genes could be confidently quantified for their allelic translational status (see Materials and Methods). Based on polysome profiling results, these 54 genes could be categorized into three groups, that is no allelic bias ($n = 35$), bias towards C57BL/6J ($n = 10$) or SPRET/EiJ allele ($n = 9$). As shown in Figs 2D and EV3B, the allelic biases at the protein level quantified by MS were on average coherent with that based on polysome data, and the MS estimates were significantly different among all the three groups.

### Cis-regulatory elements proximal to start codons contributed to ADTE

The ADTE observed in the F1 hybrid should only reflect the impact of the allelic differences in *cis*-regulatory elements residing within the transcripts. To study the potential *cis*-features accounting for the observed allelic translational bias, we first calculated the density of sequence variants between the two parental genomes for 634 genes with significant ADTE and 1,291 control genes without ADTE (restricted to single-isoform genes with unambiguous 5′/3′ UTR annotation, see Materials and Methods). As shown in Fig 3A, the genes with significant ADTE contained significantly higher density

of sequence variants than the control genes ($P = 1.7 \times 10^{-5}$, Kolmogorov–Smirnov test; see also Fig EV4A for ADTE genes with allelic TE divergence > 1.5 instead of 2.0, and Fig EV5A for ADTE genes determined based on the ribosome footprinting data).

Next, we sought to explore how these sequence variants were distributed in different genic regions. For this purpose, each gene was separated into 5′UTR, CDS and 3′UTR regions, and SNP density was calculated in each region and then normalized against the overall SNP density of the same gene. Compared to the 1,291 control genes, the 634 genes with significant ADTE showed relatively higher enrichment of SNPs in 5′UTR (Fig 3B; see also Fig EV4B for ADTE genes with allelic TE divergence > 1.5 and Fig EV5B for ADTE genes determined based on the ribosome footprinting data). Inspired by this observation, we further examined the SNP enrichment inside 5′UTR proximal to the start codon (see Materials and Methods). As shown in Fig 3C, compared with the control genes, the genes with significant ADTE exhibited on average higher SNP enrichment in the region proximal to the start codon (see also Fig EV4C for ADTE genes with allelic TE divergence > 1.5 and Fig EV5C for ADTE genes determined based on the ribosome footprinting data).

To dissect potential *cis*-elements close to the start codon accounting for the observed ADTE, given the well-known importance of Kozak sequence in translational regulation (Kozak, 1986, 1987), we first focused on the variants residing in the Kozak sequence (positions from −6 to +5 relative to start codon). Among the 634 genes with significant ADTE, 7.1% contained at least one SNP in the region, compared to 7.3% of the control genes. There was no significant difference between the two gene groups ($P = 0.93$, Fisher's exact test). It has been reported that the third nucleotide upstream of the start codon (position −3) has a dominant effect, where a purine (A or G) is important for achieving optimal TE. Consistent with the importance of this position, we found a purine in ~87% of the genes examined for ADTE. Only four genes contained transversion SNPs (purine to pyrimidine or vice versa) at this position, which did not allow any statistical analysis on the contribution of SNPs at this position to overall ADTE. Nevertheless, interestingly, among the genes with sequence variants in other positions of the Kozak sequence, we found those with a C at the position −3 tended to more frequently show significant ADTE (odds ratio = 3.02, $P = 0.059$, Fisher's exact test).

mRNA secondary structure around the start codon has been reported to affect TE (Kudla *et al*, 2009; Dvir *et al*, 2013). We therefore compared the minimum free energy (MFE) of mRNA segments (of length 20–50 nt) surrounding the start codon between the two alleles (see Materials and Methods) and correlated such difference to the observed ADTE. By large, the alleles with less stable local secondary structure around the start codon were more likely to show higher TE (Fig 3D; see also Fig EV5D for ADTE calculated based on the ribosome footprinting data).

Another category of known regulatory elements in 5′UTR includes uORFs and uAUGs (Mueller & Hinnebusch, 1986; Matsui *et al*, 2007; Calvo, 2009). Here we defined uORFs as ORFs that resided completely within the 5′UTRs, and uAUGs as AUG codons in 5′UTR but without any in-frame stop codons upstream to the start codons of main ORFs. To check whether allelic difference in the presence of uORFs or uAUGs contributed to the observed allelic TE bias, we first separated 1,640 (695) genes with uORFs (uAUGs) into two groups, one group containing 1,597 (618) genes

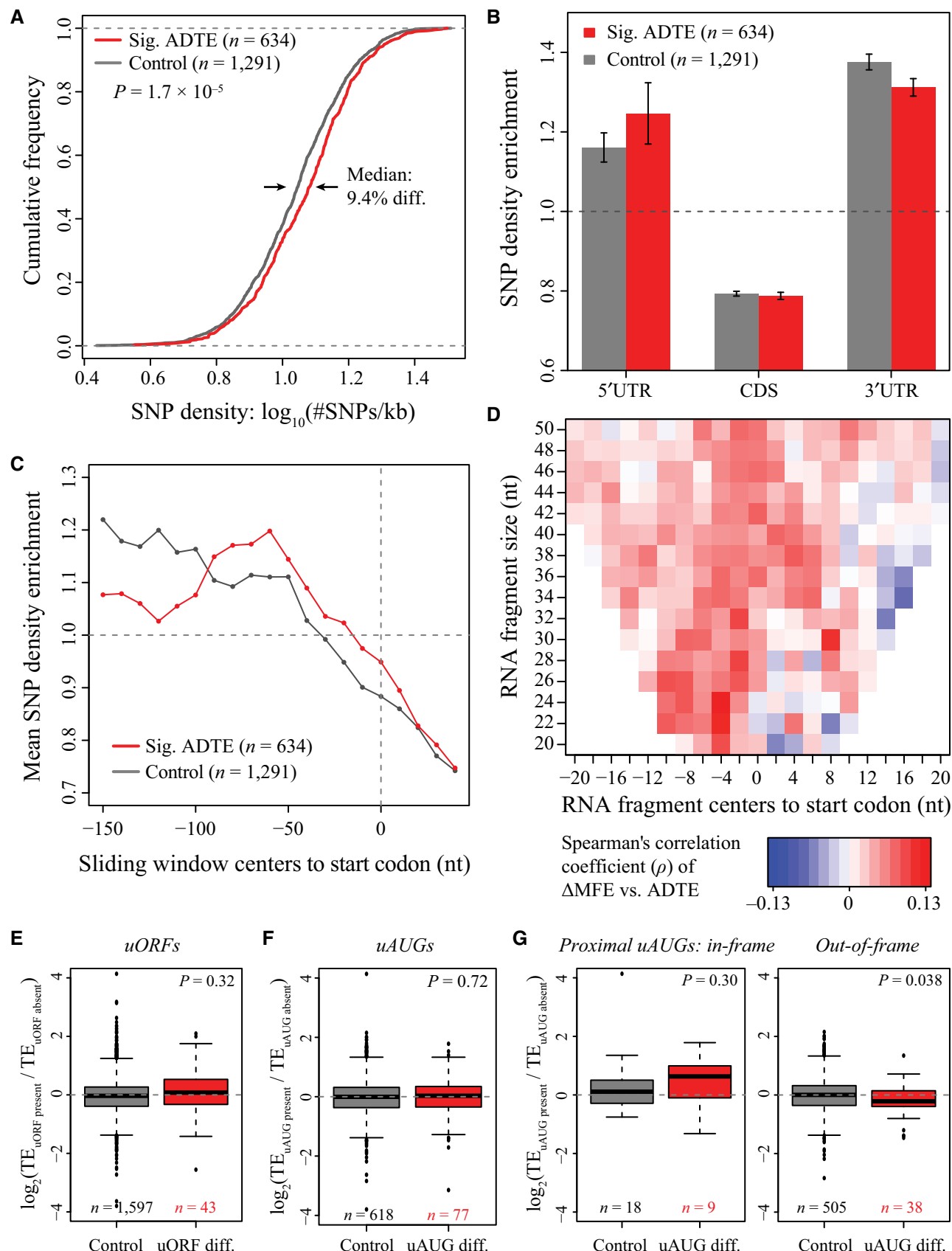

**Figure 3.**

**Figure 3.  Sequence features that were correlated with ADTE.**

A   The cumulative distribution function (CDF) of SNP density (number of SNPs per kb) for genes with significant ADTE (red) and without (control genes, grey). Compared to the control genes, the genes with significant ADTE showed significantly higher SNP density ($P = 1.7 \times 10^{-5}$, Kolmogorov–Smirnov test), with the median SNP density for the genes with significant ADTE being approximately 9.4% higher than that for the control genes.

B   Barplots showing the regional SNP density enrichment for the genes with significant ADTE (red) and the control genes (grey). Each gene was separated into 5′UTR, CDS and 3′UTR regions, and SNP density was calculated in each region and then normalized against the overall SNP density of the same gene. Compared to the 1,291 control genes, the 634 genes with significant ADTE tended to show relatively higher enrichment of SNPs in 5′UTR. Grey and red bars represent mean, and error bars represent s.e.m.

C   SNP density enrichment in 5′UTR proximal to the start codon for the genes with significant ADTE (red) and the control genes (grey). In the region proximal to the start codon (up to 200 nt upstream), we calculated the SNP density in all 100-nt sliding windows with a step size of 20 nt and then normalized against the overall SNP density of the same gene. The distance of window centre to start codon was indicated in *x*-axis, and the mean SNP density enrichment from the two gene groups was indicated in *y*-axis. Although the SNP enrichment difference in five windows had a nominal $P < 0.05$, after Benjamini–Hochberg correction for multiple testing, no windows remained significant (adjusted $P < 0.05$) (see Materials and Methods for the statistical test).

D   Heatmap showing the Spearman's correlation coefficient (ρ) between ADTE and the allelic difference in the minimum free energy (MFE) of mRNA segments surrounding the start codon. For each of mRNA segments, its length was indicated in *y*-axis and the distance of its centre to start codon was indicated in *x*-axis. Colour keys for ρ were shown below the heatmap. Note that ρ in none of the segments achieved statistical significance (FDR < 0.05) (see Materials and Methods for the statistical test).

E   Boxplots comparing the distribution of ADTE between 1,597 genes with uORF present in both alleles (grey) and 43 genes with uORF present in only one allele (red). No significant differences between the two groups were observed ($P = 0.32$, Mann–Whitney *U*-test).

F   Boxplots comparing the distribution of ADTE between 618 genes with uAUG presence in both alleles (grey) and 77 genes with uAUG presence in only one allele (red). No significant differences between the two groups were observed ($P = 0.72$, Mann–Whitney *U*-test).

G   Boxplots comparing the distribution of ADTE between 18 (505) genes with proximal in-frame (out-of-frame) uAUG presence in both alleles (grey) and 9 (38) genes with proximal in-frame (out-of-frame) uAUG presence in only one allele (red). Whereas no significant correlation was observed between ADTE and presence/absence of the proximal (≤ 100 nt upstream of the main ORF) in-frame uAUGs ($P = 0.30$, Mann–Whitney *U*-test), for genes with proximal out-of-frame uAUGs in only one allele, ADTE significantly differed from that of genes with proximal out-of-frame uAUGs in both alleles ($P = 0.038$, Mann–Whitney *U*-test).

with uORFs (uAUGs) in both alleles, and the other 43 (77) genes with uORFs (uAUGs) in only one allele. Comparing the distribution of ADTE between the two groups, we observed no significant differences between the two groups for either uORF (Fig 3E; $P = 0.32$, Mann–Whitney *U*-test; see also Fig EV5E for ADTE calculated based on the ribosome footprinting data) or uAUG (Fig 3F; $P = 0.72$, Mann–Whitney *U*-test; see also Fig EV5F for ADTE calculated based on the ribosome footprinting data). After noting that uAUGs located in the same frame as the main ORF (in-frame) or not (out-of-frame) may play different roles in affecting translation of main ORF, we separated the genes with uAUGs into two sets, each of which containing only in-frame or out-of-frame uAUGs. Interestingly, whereas we did not observe any significant correlation between ADTE and presence/absence of the in-frame uAUGs (Fig 3G; $P = 0.30$, Mann–Whitney *U*-test; see also Fig EV5G for ADTE calculated based on the ribosome footprinting data), we found that, for genes with proximal (≤ 100-nt upstream of the main ORF) out-of-frame uAUGs in only one allele, ADTE differed with marginal significance from that of genes with proximal out-of-frame uAUGs in both alleles (Fig 3G; $P = 0.038$, Mann–Whitney *U*-test; see also Fig EV5G for ADTE calculated based on the ribosome footprinting data). The observation that ADTE on average biased towards the allele without uAUG indicates that the presence of a proximal out-of-frame uAUG could negatively affect the TE of the main ORF.

In previous studies, a number of sequence features beyond start codon have also been reported to affect translation, including GC content, codon bias (measured by codon adaptation index, CAI) and the occurrence of miRNA target sites (Sandberg *et al*, 2008; Mayr & Bartel, 2009; Santhanam *et al*, 2009; Plotkin & Kudla, 2010; Vogel *et al*, 2010). To investigate whether these features accounted for the ADTE observed in this study, we separated the genes into three sets, that is no allelic bias, bias towards C57BL/6J and SPRET/EiJ allele (see Materials and Methods), and then calculated the different features for each set, separately. As a result, we did not observe

among the three sets significant disparity of the difference between the two alleles with respect to GC content and codon bias measured by CAI (Fig EV6A and B). To estimate the contribution of miRNA, we profiled the miRNA abundance in our F1 fibroblast cells and predicted the target sites for the 20, 50 and 100 most abundant miRNAs. As shown in Fig EV6C, no significant allelic difference in the number of predicted miRNA target sites could be observed among the three gene sets.

**Comparable allelic regulation of translation versus transcription, and their coordination**

In our F1 hybrid cells, the allelic bias in protein abundance is controlled by the allele-specific regulation at transcriptional as well as translational level. To explore the relative contribution of the two processes, we first calculated allelic bias in RNA abundance, likely resulting mostly from allelic transcriptional regulation. Based on only total mRNA sequencing dataset, using the same bootstrapping strategy at the same threshold (adjusted *P*-value < 0.05 and allelic divergence > 2-fold, FDR = 4.74%, see Fig EV7), out of 7,892 genes, we identified 1,041 with significant allelic differences in mRNA abundance. As shown in Fig 4A, the proportion of genes exhibiting allelic bias at mRNA abundance or translational efficiency was similar (Fig 4A; 13.2 versus 14.1%, $P = 0.11$, Fisher's exact test; see also Fig EV8A and B for different threshold setting). In addition, the allelic difference in mRNA abundance only explained 43% of the allelic divergence in poly-mRNA abundance (Fig EV8C). Both observations suggested that allelic regulation at the two levels operated with comparable importance in determining final allelic bias in protein abundance.

Previous studies in yeast have shown that allelic translation and transcription could be regulated in a coordinated fashion; however, it is still in debate whether the regulatory effects at the two levels reinforce or compensate each other (Artieri & Fraser, 2014a; McManus *et al*, 2014; Muzzey *et al*, 2014). Here based on our

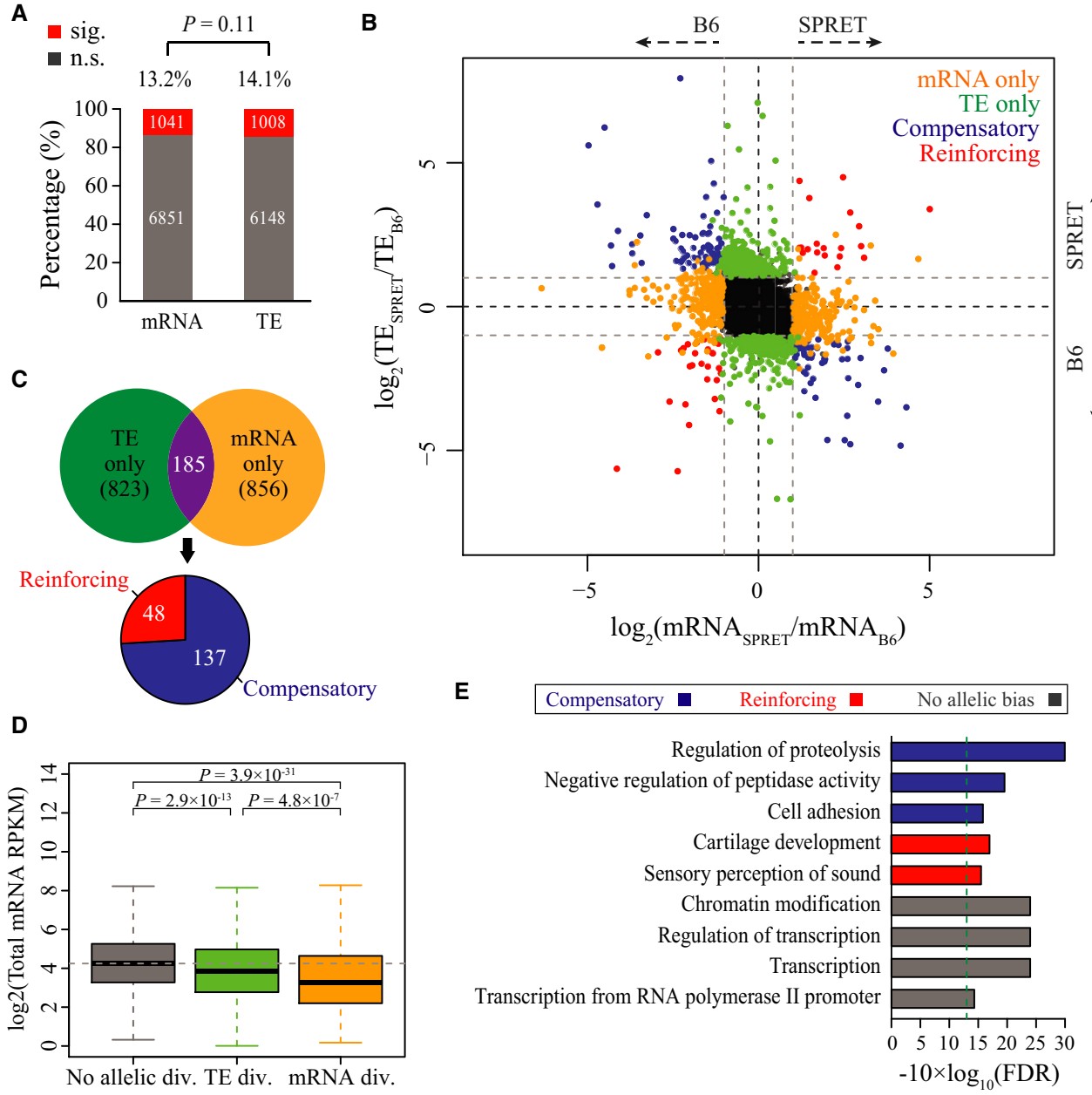

**Figure 4. Comparable allelic regulation of translation versus transcription, and their coordination.**

A   Comparable *cis*-effects at transcriptional and translational levels. Barplots showed 13.2 and 14.1% of genes with significant allelic bias at transcriptional and translational levels, respectively, and the two proportions were of no significant difference ($P = 0.11$, Fisher's exact test). Numbers of genes with or without biases at transcriptional or translational levels are indicated within the corresponding bars. See also Fig EV8A and B for the comparison at different threshold settings.

B   Scatterplot comparing each gene's allelic divergence (log2-transformed fold change) at transcriptional (*x*-axis) and translational (*y*-axis) levels. Grey dash lines indicate twofold divergence at either level. Compensatory (mRNA and TE divergent in opposite directions) and reinforcing (mRNA and TE divergent in the same direction) genes were depicted in blue and red, respectively, while genes with significant allelic bias at only mRNA level and only TE level were depicted in orange and green, respectively. See Fig EV8D and E for results at different threshold settings.

C   Venn diagram showing 185 genes exhibiting significant allelic biases at both transcriptional and translational levels, which was significantly more than that expected by chance ($P = 5.8 \times 10^{-7}$, Fisher's exact test). Among the 185 genes, 137 showed the compensatory effects between the two processes, which was approximately three times the number of genes with the reinforcing effects.

D   Boxplots showing mRNA expression levels (log2-transformed reads per kilobase per million mapped reads (RPKM) values from total mRNA sequencing data) for genes without allelic divergence at either level (grey), and genes with significant ADTE (green) or mRNA abundance (orange). On average, genes with allelic bias at either level expressed significantly lower than those without allelic bias, and genes exhibiting allelic bias at the translational level expressed significantly higher than those showing allelic bias at the transcriptional level. The *P*-values indicate the significance level of pairwise comparison of expression level among the three gene groups (Mann–Whitney *U*-test).

E   Gene Ontology (GO) enrichment of compensatory genes (blue), reinforcing genes (red) and genes without allelic bias at either level (grey). All GO terms shown are with FDR < 0.05.

dataset, we sought to address in a mammalian system whether and how the extensive allelic translational regulation coordinated with the allelic mRNA abundance. As shown in Fig 4B, 1,041 and 1,008 showed significant allelic biases in either RNA abundance or translational efficiency, respectively. Among them, 185 were exhibiting allelic biases at both levels, which was slightly but statistically significantly more than expected by chance (Fig 4C, $P = 5.8 \times 10^{-7}$, Fisher's exact test). Among these 185 overlapping genes, 137 showed the compensatory effects between the two processes (mRNA and TE divergent in opposite direction), nearly two times more frequent than those with the reinforcing effects (mRNA and TE divergent in the same direction) ($n = 48$) (Fig 4C; see also Fig EV8D and E for different threshold setting).

We then categorized the 7,892 genes into three groups based on their allelic bias at transcriptional and translational levels. Interestingly, we found that, on average, genes with allelic bias at either level expressed significantly lower than those without allelic bias, and genes exhibiting allelic bias at the translational level expressed significantly higher than those showing allelic bias at the transcriptional level (Fig 4D). Finally, we asked whether genes with or without allelic bias in transcriptional and/or translational regulation had distinct functions (see Materials and Methods). As shown in Fig 4E, the genes without allelic biases in either process were enriched in constitutive cellular processes, such as chromatin modification, and transcription. While compensatory genes also showed enrichment of some essential functions, such as regulation of proteolysis, reinforcing genes were enriched in two specific functional categories, that is cartilage development and sensory perception of sound.

## Discussion

Changes in translational efficiency play an important role in shaping phenotypic diversity during evolution. To globally investigate *cis*-divergence in translational regulation in mammals, we performed a first genome-wide survey of allele-specific translational regulation in a hybrid mouse system. Our data demonstrated that *cis*-divergence in translation and transcription was of comparable importance in determining allelic bias in protein abundance, and the *cis*-divergence in the two regulatory precesses more often buffered than enhanced each other. The large set of genes with *cis*-divergent translational regulation collected in this study also enabled to systematically characterize the potential *cis*-elements in translational regulation.

To identify the genetic variants with regulatory effects on gene expression, including ones lying in *cis*, a frequently used method is eQTL mapping, in which different genotypes are correlated with gene expression levels in a large population with diverse genetic backgrounds (Pickrell *et al*, 2010; Majewski & Pastinen, 2011; Lappalainen *et al*, 2013). Recently, this strategy has been extended to study the genetic regulation on protein abundance (Ghazalpour *et al*, 2011; Skelly *et al*, 2013; Wu *et al*, 2013; Battle *et al*, 2015). However, since the deep proteomic analysis of a large number of samples is challenging, many true pQTLs probably escaped the detection, especially those with smaller effect size (Brem & Kruglyak, 2005). An alternative approach that could more directly address the effect of *cis*-divergence is to analyse the allelic difference in F1 hybrids between two distantly related parental strains. Very

recently, a couple of studies have used this strategy in hybrid yeast to characterize allele-specific TE based on deep sequencing-based ribosome footprinting (Albert *et al*, 2014; Artieri & Fraser, 2014b; McManus *et al*, 2014). In this study, we applied the same approach in mice and chose the F1 hybrid between C57BL/6J and SPRET/EiJ inbred strains as our model. Among all the mouse strains with high-quality genome assembly, SPRET/EiJ has the largest number of sequence variants relative to C57BL/6J. This large genomic divergence first provides a large number of potential regulatory variants between the two strains. Second, more importantly, it allows the use of deep sequencing or MS-based approaches to distinguish RNA transcripts or peptides derived from either allele. Here, we used deep sequencing-based polysome profiling and ribosome footprinting as well as MS coupled with pulse labelling of newly synthesized protein to measure allelic TE. As expected, given the much lower number of peptides that could be detected and assigned to either allele, MS-based approach could only be used to identify the allelic bias in tens of different genes. Deep sequencing-based ribosome profiling and polysome profiling data serve as a close proxy to mRNA translation status, although both only capture a snapshot of ribosome–RNA association without taking translational elongation into account. With much higher sensitivity, they are however more useful in quantifying allele-specific TE. Compared to ribosome footprinting, which measures the number of ribosomes associated with individual mRNA transcripts, polysome profiling quantifies the proportion of cellular mRNAs associated with polyribosome and therefore yields lower resolution regarding the number of associated ribosomes. Nevertheless, while ribosome profiling captures RPFs of only ~29–33 nt in length, polysome profiling can be used to generate longer sequencing reads that more likely cover at least one sequence variants between the two alleles. Empirically, polysome profiling with paired-end $2 \times 100$ nt sequencing resulted in > 60% reads with unambiguous allelic origin assigned, compared to 19% reads from ribosome profiling (Table EV1, Figs 1B and EV2B). Moreover, the uncertainty in ADTE estimation (i.e. bootstrapping standard deviation) based on polysome profiling was much lower than that based on ribosome footprinting (Fig EV2D). After considering all the pros and cons of both approaches, we decided to base our analysis on polysome profiling data. In this way, 1,008 genes were identified with significant ADTE, of which 688 had sufficient allelic RFP data. Among them, 460 (66.9%) showed also significant allelic bias. Importantly, no single gene showed significant allelic TE divergence in the opposite direction between polysome profiling and ribosome footprinting data. Finally, the allelic bias of 54 genes estimated using MS data were on average coherent with that based on polysome data, again validating that our polysome profiling data could be reliably used for quantifying ADTE.

Compared to the genes without ADTE, the genes with allelic biases contained higher SNP density, suggesting that the divergent TE identified in our study was indeed caused by *cis*-variants. In addition, we observed that SNPs associated with TE divergence were more enriched in 5′UTR, particularly in the vicinity of start codon, indicating that the genetic variants accounting for the observed ADTE preferentially functioned by affecting translation initiation, which was in great accordance with previous findings that translation is predominantly regulated at the initiation stage (Jackson *et al*, 2010). Kozak sequence is one of the most well-known elements in controlling translation initiation, and within the

Kozak sequence, it has been reported that there is a strong preference of a purine at the third nucleotide upstream of the start codon (position $-3$) for efficient translation (Kozak, 2005). Interestingly, whereas ~87% genes analysed in this study contained a purine and nearly no genes had transversion SNPs at this position, among the genes with SNPs in other positions of the Kozak sequence, we found those with a C at the position $-3$ tended to more frequently exhibit significant ADTE. This suggests that sequence variants at other positions of the Kozak element could affect translation more likely under non-optimal context. Beyond Kozak element, we observed the variants affecting local secondary structure surrounding the start codon could result in ADTE, largely agreeing with the previous findings in yeast (Dvir et al, 2013; Shah et al, 2013; Artieri & Fraser, 2014b; Muzzey et al, 2014). Recently, based on genome-wide analysis of RNA secondary structure, it has been observed in *Saccharomyces cerevisiae*, *Arabidopsis thaliana* and human that RNA fragments in the vicinity of start codons tend not to form stable secondary structure (Kertesz et al, 2010; Wan et al, 2012, 2014; Ding et al, 2014). A slight alteration of this local secondary structure due to a single SNP might be sufficient to cause detectable difference in TE. It has also been reported that the presence of uORFs or uAUGs would decrease TE (Mueller & Hinnebusch, 1986; Vattem & Wek, 2004; Matsui et al, 2007; Calvo, 2009), yet in this study, the difference (presence/absence) of overall uORFs or uAUGs between the two alleles appeared not to correlate with the observed ADTE. However, after separating the proximal uAUG into in-frame and out-of-frame categories, we observed that out-of-frame proximal uAUGs could negatively affect TE. This observation agrees with empirical evidences that uAUGs would diminish the translation of main ORFs by reducing the number of ribosomes reaching the downstream AUGs. Such negative impact was not observed for in-frame uAUGs, possibly due to the fact that, in contrast to out-of-frame uAUGs, the in-frame uAUGs could generate N-terminal extended protein isoforms (Kozak, 2005; Medenbach et al, 2011; Dvir et al, 2013). Indeed, Dvir et al (2013) demonstrated with their reporter assays that translation from the main ORF was efficiently attenuated by only the out-of-frame uAUGs, but not in-frame uAUGs.

Surprisingly, we did not find the significant impact of several known *cis*-regulatory features in determining ADTE observed in this study, such as the number of miRNA binding sites and codon bias. Possible explanations include, first, ADTE might be due to the combined effect of a large set of diverse mechanisms and the contribution of individual feature with smaller effect sizes might not be sufficient to reach statistical significance. In fact, miRNA binding reduces protein output through mRNA degradation and translational repression (Bartel, 2009), and in many recent studies, it only shows modest influence in TE (Guo et al, 2010; Mukherji et al, 2011; Eichhorn et al, 2014). In addition, the presence of these sequence features, such as miRNA target sites, could be predicted only with limited accuracy. It has been reported that at most 60–70% computationally predicted miRNA target sites are functionally relevant (Lewis et al, 2003; Selbach et al, 2008). Besides, codon optimality has been hypothesized to exert its effect mostly by modulating translational elongation rate. It might not change mRNA–ribosome association and thus not susceptible to our ADTE measurement based on polysome profiling (Tuller et al, 2010; Novoa & Ribas de Pouplana, 2012; Presnyak et al, 2015). Moreover, several previous

ribosomal profiling studies also failed to detect codon-specific differences in the translation of optimal and non-optimal codons (Ingolia et al, 2009; Qian et al, 2012; Charneski & Hurst, 2013), indicating this hypothesis is still largely in debate. Finally, we based our analysis on the RefSeq annotation while our F1 fibroblast cells might express alternative isoforms. Indeed, it is conceivable that differential usage of alternative 5′ and/or 3′UTRs, and even the alternative start/stop codons, between the two alleles could lead to allelic difference in TE. Global dissection of such effects awaits future studies, where experimental data on allele-specific isoform usage will need to be collected.

It has been shown previously that across species, protein levels are less diverged than mRNA abundances (Garge et al, 2010; Khan et al, 2013; Wu et al, 2013; Hause et al, 2014). One possible mechanism is that divergence in translational and transcriptional regulation offsets each other. Consistent with this, in our F1 hybrid system, among the genes with allelic bias at both translation and transcription level, almost three-fourths showed the compensatory effects between the two processes, that is mRNA and TE divergent in opposite directions. However, the number of genes with compensatory effect that we identified here was relatively small; therefore, the offsetting effect between the two regulatory processes alone may not be sufficient to explain the attenuated protein divergence. Indeed, a recent study, based on their genome-wide eQTL, rQTL and pQTL mapping data, suggested that protein degradation might play a more important role in maintaining robust protein cellular abundance during evolution (Battle et al, 2015).

Genes with or without allelic bias in transcriptional and/or translational regulation were on average of different mRNA abundance and enriched with distinct functional categories. First, as expected, genes without allelic difference in either mRNA abundance or protein synthesis, which are likely under the high selection pressure, expressed at higher level and were enriched in housekeeping functions. Second, the observation of higher mRNA level for genes regulated at translational level than those at transcriptional level makes intuitive sense since translation regulation requires higher amount of regulatory substrates, that is existing mRNAs. Finally, while the genes with compensatory effect between allelic regulations at the two levels were enriched in more essential functions, the genes whose allelic difference in protein abundance was amplified by both transcriptional and translational processes were enriched in specific functional categories. Whether this gene group could explain in part the phenotypic difference between the two mouse strains awaits future studies of physiologically more relevant tissues.

## Materials and Methods

### F1 hybrid mouse fibroblast cell cultures

The F1 hybrid mice were obtained as described before (Gao et al, 2013). Adult mouse fibroblast cells were isolated and cultured according to the protocol from ENCODE project (http://genome.ucsc.edu/ENCODE/protocols/cell/mouse/Fibroblast_Stam_protocol.pdf) with modification of cell culture medium (RPMI 1640 medium, GlutaMAX™ supplement with 10% FBS and 1% penicillin/streptomycin).

## mRNA sequencing

Total RNAs from mouse fibroblast cells were extracted using TRIzol reagent (Life Technologies) following the manufacturer's protocol. Truseq Stranded mRNA sequencing libraries were prepared with 500 ng total RNA according to the manufacturer's protocol (Illumina). The libraries were sequenced in $2 \times 100$ nt manner on HiSeq 2000 platform (Illumina).

## Polysome profiling

Mouse fibroblast cells were grown to 80% confluency. Prior to lysis, cells were treated with cycloheximide (100 μg/ml) for 10 min at 37°C. Then, cells were washed with ice-cold PBS (supplemented with 100 μg/ml cycloheximide) and further lysed in 300 μl of lysis buffer (10 mM HEPES pH 7.4, 150 mM KCl, 10 mM MgCl$_2$, 1% NP-40, 0.5 mM DTT, 100 μg/ml cycloheximide). After lysing the cells by passing eight times through 26-gauge needle, the nuclei and the membrane debris were removed by centrifugation (15,682 $g$, 10 min, 4°C). The supernatant was then layered onto a 10-ml linear sucrose gradient (10–50% [w/v], supplemented with 10 mM HEPES pH 7.4, 150 mM KCl, 10 mM MgCl$_2$, 0.5 mM DTT, 100 μg/ml cycloheximide) and centrifuged (160,000 $g$, 120 min, 4°C) in an SW41Ti rotor (Beckman). Fractions were collected and digested with 200 μg proteinase K in 1% SDS and for 30 min at 42°C. RNA from polysome fractions were recovered by extraction with an equal volume of acid phenol–chloroform (pH 4.5), followed by ethanol precipitation. TruSeq Stranded Total RNA libraries were prepared with 500 ng RNA according to the manufacturer's protocol (Illumina). The libraries were sequenced in $2 \times 100$ nt manner on HiSeq 2000 platform (Illumina).

## Ribosome profiling

Mouse fibroblast cells were cultured and lysed in the same way as for polysome profiling (see above). After lysis, ribosome-protected fragments were collected as described in Ingolia *et al* (2012), with minor modifications. In brief, cell lysate was treated with RNase I at room temperature for 45 min. The nuclease digestion was stopped by adding SUPERase In™ RNase inhibitor (Invitrogen) and then loaded onto a linear sucrose gradient (10–50%). After ultracentrifugation, monoribosome was recovered and RNA was isolated as described for polysome profiling (see above). rRNA was removed using Ribo-Zero™ Magnetic Kit (Human/Mouse/Rat) (Epicentre). The 28- to 32-nt ribosome-protected fragments were purified through 15% (wt/vol) polyacrylamide TBE-urea gel. The size-selected RNA was end-repaired by T4 PNK for 1 h at 37°C. The sequencing libraries were then generated using TruSeq Small RNA Sample Preparation kit (Illumina) and sequenced in $1 \times 50$ nt manner on Illumina HiSeq 2000 platform.

## Reference sequences and gene annotation

The reference sequences of the C57BL/6J genome were downloaded from the Ensembl FTP server (ftp://ftp.ensembl.org/pub/release-72/fasta/mus_musculus/dna/; version GRCm38, Release 72). The Ensembl gene annotation of C57BL/6J was also downloaded from the Ensembl FTP server (ftp://ftp.ensembl.org/pub/release-72/

gtf/mus_musculus; Release 72). The RefSeq gene annotation was downloaded from the UCSC genome browser (http://hgdownload.soe.ucsc.edu/goldenPath/mm10/database/) on 5 June 2014. The SNPs and indels between C57BL/6J and SPRET/EiJ were downloaded from the Sanger Institute (ftp://ftp-mouse.sanger.ac.uk/; Release v3, Build 137). The vcf2diploid tool (version 0.2.6) in the AlleleSeq pipeline was used to construct the SPRET/EiJ genome by incorporating the SNPs and indels into the C57BL/6J genome (Rozowsky *et al*, 2011). The chain file between the two genomes was also reported as an output, which was further used with the UCSC liftOver tool. The liftOver tool from the UCSC Genome Browser (Kuhn *et al*, 2013) was applied to get the SPRET/EiJ gene annotation. Given the genome sequences and the gene annotation, transcriptome reference sequences of both strains were built using custom Perl scripts.

## Allele-specific sequencing read mapping

The sequencing reads were first subjected to adapter removal using flexbar with the following parameters: -u 3 -m 32 -ae RIGHT -at 3 -ao 1 (Dodt *et al*, 2012). Read pairs that were concordantly mapped to the reference sequences of rRNA, tRNA, snRNA, snoRNA and miscRNAs (available from Ensembl and RepeatMasker annotation) using Bowtie2 (version 2.1.0) (Langmead & Salzberg, 2012) with default parameters (in –end-to-end & –sensitive mode) were excluded. The remaining reads were then mapped to the both C57BL/6J and SPRET/EiJ transcriptome reference sequences using Bowtie2 (version 2.1.0) with the same parameters as above but allowing no more than four mismatches per read pair. Concordantly mapped read pairs (i.e. mates of a read pair mapped to the same transcript with opposite orientation) were then assigned to the C57BL/6J or SPRET/EiJ allele with less mapping edit distance; read pairs with equal edit distance to either allele were assigned as "common". Read pairs that mapped to sexual chromosomes and mitochondrial DNA were excluded for further analysis.

## Filtering of SNP loci with potential allelic read mapping biases

To estimate ADTE, only the reads that could be unambiguously assigned to SNP loci from either allele were counted (see below). Due to potentially incomplete annotation of SNPs at paralogous gene or pseudogenes in the SPRET/EiJ genome, some reads, which could be mapped to multiple gene loci if the C57BL/6J sequences used as a reference, were mapped to a unique position in the SPRET/EiJ allele. In such cases, removal of multiple mapped reads (only from C57BL/6J allele) could lead to inaccurate calculation of ADTE. Therefore, we filtered out SNP loci if: (i) more multiple mapped reads than uniquely mapped reads were aligned at the loci from either allele, and (ii) the ratio of allelic abundance at the loci calculated based on multiple mapped reads differs by > 1.5-fold from that using uniquely mapped reads.

## Estimation of Allelic Divergence in Translational Efficiency (ADTE)

After SNP loci filtering (see above), protein-coding genes with ORF containing at least five SNPs in constitutive exons supported by sufficient allelic reads (i.e. for a SNP locus, $Poly_{SPRET} + Poly_{B6} \geq 10$ and $mRNA_{SPRET} + mRNA_{B6} \geq 10$ and $mRNA_{SPRET} + Poly_{SPRET} \geq 10$

and $mRNA_{B6} + Poly_{B6} \geq 10$, where $Poly_{SPRET/B6}$ and $mRNA_{SPRET/B6}$ represent the number of poly-mRNA and total mRNA reads aligned to the SPRET/EiJ or C57BL/6J allele at the SNP locus, respectively) were subjected to ADTE estimation using the following formula:

$$ADTE = \log_2 \left\{ \left( \sum Poly_{SPRET} / \sum Poly_{B6} \right) \right.$$
$$\left. / \left( \sum mRNA_{SPRET} / \sum mRNA_{B6} \right) \right\}$$

where $\Sigma$ represents the sum of allelic reads from all the SNP loci belonging to the same ORF.

Similar as described in Muzzey *et al* (2014), a bootstrapping procedure was applied to assess the estimation uncertainty. In short, for each ORF consisting of a list of $n$ ($n \geq 5$) SNP loci, we generated 5,000 new lists, each consisting of $n$ SNP loci that were chosen at random with replacement from the original list. For each of the 5,000 random list, ADTE was calculated and then yielded a bootstrap distribution, from which we got the bootstrapping mean and standard deviation, as shown in Fig 2A. Non-zero bootstrapping means indicated the TE of the two alleles was not equal. To determine the statistical significance of genes with ADTE, we calculated a *P*-value based on the *Z*-score that represented how many folds of standard deviation the bootstrapping mean deviated from zero. The raw *P*-values were then adjusted using the Benjamini–Hochberg method. To determine the false discovery rate based on our experimental replicates, we applied a similar permutation strategy as that used in Sterne-Weiler *et al* (2013). In short, gene labels were shuffled for 100 times in both replicates, and in each of the 100 shuffled sets, we counted the number of genes in both replicates meeting the fold change (FC) requirement ($|FC| > x$) and bootstrapping significance requirement (adjusted *P*-value < 0.05), and biased towards the same allele, denoted as FP($x$). Then, the FDR in each set for each value of $x$ was estimated as FP($x$) divided by the number of real genes passing the same criteria. We applied the same bootstrapping procedure to assess the uncertainty of ADTE estimated based on ribosome footprinting data. The ADTE calculation was executed in R version 3.1.1 (R Core Team, R Foundation for Statistical Computing, Vienna, Austria; http://www.R-project.org) with custom scripts.

### PacBio sequencing and data analysis

Starting from 500 ng total RNA or polysomal RNA, reverse transcription (RT) was performed using random hexamer and SuperScript II reverse transcriptase. PCR was followed using 1 µl of RT product as template in 50 µl of Phusion High-Fidelity DNA Polymerase system (NEB). PCR primers were designed for amplifying the genic region covering $\geq 3$ sequence variants between C57BL/6J and SPRET/EiJ transcripts. PCR program was as follows: 30 s at 98°C, followed by 30 cycles of 10 s at 98°C, 30 s at 60 °C and 45 s at 72°C, and a final elongation of 5 min at 72°C. The amplified RT–PCR products from total RNA or polysomal RNA were mixed separately. The mixed products were then purified using Agencourt AMPure XP system (Beckman Coulter) and quantified by Qubit HS dsDNA measurement system (Life Technology). These mixed PCR products were sequenced on PacBio RS SMRT platform according to the manufacturer's instruction. All the primer sequences were listed in Table EV2.

Sequence reads from the PacBio RS SMRT chip were processed through PacBio's SMRT-Portal analysis suite to generate circular consensus sequences (CCSs). The CCSs were then mapped to both alleles of target genes using BLASR (part of SMRT analysis, version 2.2.0) with default parameters except -minReadLength 300. The CSS reads were assigned to C57BL/6J or SPRET/EiJ allele with fewer mismatches. The numbers of reads assigned to either allele of each gene from total mRNA and poly-mRNA were counted and used to calculate ADTE.

### Azidohomoalanine (AHA) pulse-labelled samples

A similar approach as applied in Khan *et al* (2012) was used to measure allele-specific protein abundance. Parental SPRET/EiJ, C57BL/6J and the F1 fibroblasts were cultured in stable isotope labelling by amino acids in cell culture (SILAC) DMEM (Life Technologies) (supplemented with 10% dialysed FBS (Sigma-Aldrich) and 1% penicillin/streptomycin) containing either standard or heavy versions of lysine [light Lys-0 (L), medium Lys-4 (M) or heavy Lys-8 (M)] and arginine [light Arg-0 (L), medium Arg-6 (M) or heavy Arg-10 (H)] (Ong *et al*, 2002). In this way, we fully labelled the SPRET/EiJ-derived proteins light, C57BL/6J proteins medium and F1 heavy. Cells were washed twice in pre-warmed PBS before being depleted from methionine in DMEM lacking methionine (Sigma-Aldrich) (supplemented with 10% dialysed FBS (Sigma-Aldrich) and 1% penicillin/streptomycin) for 90 min. The cells were then pulsed with 1 mM of the methionine surrogate azidohomoalanine (AHA, Anaspec) for 90 min. During the pulse, newly synthesized proteins incorporate the unnatural amino acid containing an azido group. The azido group is subsequently used to covalently link the nascent proteins to alkyne-bearing agarose beads by click chemistry (Dieterich *et al*, 2006). AHA-labelled cells were scraped in ice-cold PBS and snap-frozen. Cell lysis, click reaction between AHA-containing newly synthesized proteins and alkyne agarose beads, reduction and alkylation was performed according to the protocol of the "click-it protein enrichment kit" (Sigma). Beads were washed sequentially in SDS buffer [1% SDS, 100 mM Tris, 250 mM NaCl, 5 mM EDTA (pH 8) (Sigma)], 8 M urea in 100 mM Tris (pH 8) and 80% acetonitrile before finally being washed in 50 mM ammonium bicarbonate (pH 8). Proteins were digested "on bead" first by lysyl endopeptidase (LysC, Wako chemicals) before being trypsinated (Trypsin, Promega) overnight. Digested peptides were acidified by adding trifluoroacetic acid and then stored on C18 Stage-Tips (Rappsilber *et al*, 2003). To clean the sample from polymers, which easily accumulate during the sample preparation, peptides were eluted from the StageTips with 80% acetonitrile and 0.5% acetic acid (Buffer B), vacuum-dried and resuspended in no-salt buffer (0.5% formic acid and 15% acetonitrile) before being put on strong cation exchange (SCX) tips (Empore, 3M). SCX tips were washed in no-salt buffer before peptides were eluted by adding high-salt buffer (0.5% formic acid, 15% acetonitrile and 500 mM ammonium acetate). Peptides were put back on StageTips and desalted with Buffer A (5% acetonitrile and 0.1% formic acid).

### Mass spectrometry (MS)

Samples were eluted from StageTips by Buffer B. Acetonitrile was evaporated using a speed vac and samples resuspended in Buffer A.

Peptides were separated on a 2,000 mm monolithic column with a 100-μm inner diameter that were kindly provided by Yasushi Ishihama (Kyoto University). We applied an 8-h gradient of increasing acetonitrile concentration with a flow rate of 300 nl/min on a nLC 1000 HPLC system (ThermoScientific). In addition, peptides were separated on a 150-mm column with 75-μm inner diameter packed in-house with ReproSil-Pur 120 C18-AQ 3-μm resin (Dr. Maisch GmbH) using 4-h gradients and 250 nl/min flow rate. An electro-spray ion source (ThermoScientific) was used to ionize the peptides that were subsequently analysed using a Q-Exactive mass spectro-meter (ThermoScientific). The system was run in data-dependent mode selecting the 10 most abundant ions for fractionation for higher energy collision-induced dissociation. The full scans were performed with a resolution of 70,000, a target value of 3,000,000 ions and a maximum injection time of 20 ms. The MS/MS scans were performed with a 17,500 resolution, a 1,000,000 target value and a 60-ms maximum injection time.

## MS data analysis

Raw files were analysed using MaxQuant v1.5.1.2 (Cox & Mann, 2008) using default settings. MS/MS spectra were searched against two *in silico* digested databases created from the 1-frame translated ORFs of B6 and SPRET with common contaminants added. This way all proteins were present in two forms during the search: one from the B6 and one from SPRET database distinguishable by amino acid changes caused by the non-synonymous SNPs. In parallel, the MS/MS spectra were searched against a reversed version of the two databases to control the false discovery rate that was set to 1% at both the peptide and protein levels. C-termi-nal carbamidomethylation was set as a fixed modification while acetylation of protein N-termini and methionine oxidation were set as variable modifications. Lys4 and Arg6 were set as medium labels and Lys8 and Arg10 were set as heavy labels. Trypsin/P was set as the protease and "match between runs" was activated. We used reQuantified values only for the "SNP peptides" to retrieve peptide ratios, but more accurate non-reQuantified values for "shared peptides" to obtain quantifications on the protein level.

For downstream analysis, the non-normalized peptide ratios (peptides.txt output of MaxQuant) were used. This was necessary since MaxQuant only reports SILAC ratios in the evidence.txt for all label combinations (H/L, H/M, M/L) when peptides are detected in all three SILAC states. Peptides were grouped into proteins according to the MaxQuant protein identifications. Peptides identified in both the B6 database and the SPRET data-base (shared peptides) were combined with peptides identified in only one of the two databases (SNP peptides). Shared peptides should have all three SILAC labels present (L (SPRET), M (B6) and H (F1)) while SNP peptides should have only two SILAC labels present (either H (F1) and L (SPRET) or H (F1) and M (B6)). The allele-specific expression was calculated based on the difference in the abundance between the shared peptides and the SNP peptides as follows:

$$\text{SPRET allele } [\%] = \frac{\text{median}\left(\frac{\text{H}}{\text{L}} \text{ ratio}\right)_{\text{SNP peptides}}}{\text{median}\left(\frac{\text{H}}{\text{L}} \text{ ratio}\right)_{\text{shared peptides}}} \times 100$$

$$\text{B6 allele } [\%] = \frac{\text{median}\left(\frac{\text{H}}{\text{M}} \text{ ratio}\right)_{\text{SNP peptides}}}{\text{median}\left(\frac{\text{H}}{\text{M}} \text{ ratio}\right)_{\text{shared peptides}}} \times 100.$$

Peptide ratios from the peptide.txt were weighted against their ratio counts before the median was taken for the allele-specific protein expression calculation. All analysis on MaxQuant output was performed by R version 3.1.1 (R Foundation for Statistical Comput-ing, Vienna, Austria). Out of the proteins detected with shared and allele-specific peptides ($n = 737$), we retained only proteins with 1 or more MS/MS counts of both SNP and shared peptides ($n = 168$) (Cox & Mann, 2008). Summed percentages of B6 and SPRET alleles were calculated for each protein. Proteins were finally filtered to be within the range of a summed percentage of $100 \pm 20\%$ ($n = 54$).

## miRNA profiling

Total RNAs from mouse fibroblast cells were extracted using TRIzol reagent (Life Technologies) following the manufacturer's protocol. Small RNA sequencing libraries were prepared with 1 μg total RNA using TruSeq Small RNA Sample Preparation Kit (Illumina), as described before (Li *et al*, 2013). The libraries were sequenced in $1 \times 50$ nt manner on HiSeq 2000 platform (Illumina).

## SNP density enrichment

Genes were first divided into 5′UTR, CDS and 3′UTR according to RefSeq annotation, and SNP density was calculated in each of the three regions, and then divided by the overall SNP density of the same genes, to eliminate the effect of difference in overall SNP density between different gene groups. Genes with multiple isoforms or without 5′UTR/3′UTR annotation in RefSeq were excluded in this analysis, which resulted in 634 genes with significant ADTE and 1,291 control genes (allelic TE fold change < 1.2 in both replicates).

To further examine the 5′UTR region, particularly the region close to the start codon, for each of the 634 and 1,291 genes, within the 5′UTR proximal to the start codon (up to upstream 200 nt), we calculated enrichment of SNP density (over the SNP density of the same gene) in a 100-nt sliding window with a step size of 20 nt. To assess the significance of the difference in SNP density enrichment between the two gene groups, we permuted for 1,000 times the genes' group labels, calculated the difference in mean SNP density enrichment of the two gene groups and counted the probability reflecting how often a greater difference would be observed in the permuted datasets. The raw *P*-values were then adjusted using the Benjamini–Hochberg method.

We also performed the above analysis with increased number of ADTE genes by releasing the allelic TE divergence fold change from 2.0 to 1.5, which resulted in 1,140 genes. These results were presented in Fig EV4.

## Local RNA secondary structure

Local RNA secondary structure minimum free energy (MFE) was calculated using RNAfold from the ViennaRNA package version 2.1.9 with default parameters at a temperature of 37°C (Lorenz *et al*, 2011). We compared the MFE of mRNA segments (of length 20–50 nt) surrounding the start codon between the two alleles and

correlated such difference with ADTE. Genes without SNPs in the sliding window were excluded. To access the significance of the correlation between ADTE and MFE allelic difference, that is to determine the false discovery rate (FDR) for the observed correlation coefficients, we permuted for 1,000 times the gene labels of the ADTE and MFE values, re-calculated the correlation coefficients for each individual segment separately and then computed the FDR as the ratio between the frequency of observing in the whole permuted dataset correlation coefficients equal to or greater than a specific value, and the frequency in the real data.

**Other sequence features including miRNA binding sites, codon bias and GC content**

miRNA target sites in 3′UTR were counted using custom Perl script by matching three site types (i.e. 8mer, 7mer-m8, 7mer-1A). Codon adaptation index (CAI) of coding sequence was calculated using CodonW version 1.4.4. GC content of transcript sequences was calculated with R package seqinr (version 3.1-3). Genes with multiple isoforms in RefSeq were excluded in the analyses, and those without 5′UTR/3′UTR annotation were excluded in parts of the analysis. In the analysis of these sequence features, we further split the genes with significant ADTE into two sets, one with TE biased towards the C57BL/6J allele and the other with TE biased to the SPRET/EiJ allele. Together with the control gene sets, we tested whether there were any significant differences in each sequence feature among the three gene sets.

**Gene ontology enrichment analysis**

The gene symbols were mapped to GO terms using R packages GO.db, AnnotationDbi and org.Mm.e.g.db. The 7,156 genes examined for ADTE in this study were chosen as the background set, and GO terms with at least 10 genes from this background set were tested for enrichment in each of the three gene sets (compensatory genes, reinforcing genes, genes without allelic bias) using a hypergeometric test. To adjust the *P*-values resulting from multiple comparison due to a number of GO terms being tested, we determined a family-wise false discovery rate (FDR) by permuting gene assignments, repeating the above testing procedure for 1,000 times, and only keeping the most significant *P*-value observed in each permutation. Then, an FDR for each GO term was obtained by counting how often a *P*-value of greater significance would be observed in the permutated datasets. The GO analysis was executed in R version 3.1.1 (R Core Team, R Foundation for Statistical Computing, Vienna, Austria; http://www.R-project.org) with custom scripts.

**Data access**

All raw sequencing data from this publication have been deposited to the European Nucleotide Archive (http://www.ebi.ac.uk/ena) with the accession number ERP009292. The mass spectrometry proteomics data have been deposited to the ProteomeXchange Consortium (http://proteomecentral.proteomexchange.org) via the PRIDE partner repository with the dataset identifier PXD002337.

Expanded view for this article is available online:
http://msb.embopress.org

## Acknowledgements

We thank Claudia Quedenau, Madlen Sohn, Mirjam Feldkamp and Claudia Langnick for their excellent technical assistance; Dr. Jean Jaubert and Dr. Xavier Montagutelli from the Pasteur Institute for providing F1 hybrid mice; Dr. Yasushi Ishihama from Kyoto University for kindly providing monolithic column (GL Sciences, Japan) used in generating proteomics data; and Christian Sommer and Dr. Koshi Imami for their technical assistance. As part of the Berlin Institute for Medical Systems Biology at the MDC, the research group of Wei Chen is funded by the Federal Ministry for Education and Research (BMBF) and the Senate of Berlin, Berlin, Germany (BIMSB 0315362A, 0315362C).

## Author contributions

JH, XW and WC conceived and designed the project. JH did the experiments with help from WS. XW analysed the data. EM and HZ generated and analysed the proteomics data with advice from MS. JH, XW and WC wrote the manuscript with input from EM and HZ. All authors read and approved the final manuscript.

## Conflict of interest

The authors declare that they have no conflict of interest.

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
