## [Review Process File · Molecular Systems Biology]

Extensive allele-specific translational regulation in hybrid mice

Jingyi Hou, Xi Wang, Erik McShane, Henrik Zauber, Matthias Selbach, Wei Chen

Corresponding author: Wei Chen, Max-Delbrück-Centrum für Molekulare Medizin

Review timeline:

Submission date:	17 April 2015
Editorial Decision:	19 May 2015
Revision received:	12 June 2015
Accepted:	08 July 2015

Editor: Thomas Lemberger

Transaction Report:

1st Editorial Decision

19 May 2015

Thank you again for submitting your work to Molecular Systems Biology. We have now heard back from the two referees who agreed to evaluate your manuscript. As you will see from the reports below, the referees find the topic of your study of potential interest. They raise, however, concerns on your work, which should be convincingly addressed in revision.

Without repeating all the comments from the reports, the reviewers raise two major important points:

- it remains unclear whether multiple-testing corrections have been applied and, as such, some conclusions remain unclear.
- the possibility of spurious correlations between expression data and the calculated TE should be discarded

Reviewer #1:

This manuscript by Hou et al. addresses an important question. The range of data generated and analyzed is impressive, and it is presented in a straightforward way. My only substantive question is, was there any correction for the number of statistical tests performed? For example in Fig 3D, it appears ~250 tests were performed, so ~10-15 would be expected to reach $p < 0.05$ just by chance. Using a more stringent cutoff in this and other figures with multiple tests (e.g. Fig 3C) may help to highlight the truly significant results. By the same token, the result on out of frame uAUGs is based on a p-value of 0.038, which would not be significant after correcting for even just the four uORF/uAUG-related tests in Fig 3E-G. If these results could be replicated with the ribosome profiling data, they would be greatly strengthened; but if the same trend is not seen there, I suspect they may be false positives, in which case the claims about them made in the abstract and elsewhere could be toned down.

Reviewer #2:

In this manuscript, Hou and colleagues analyze the genetic basis of gene expression variability, incorporating translational as well as transcriptional effects, in an interspecies hybrid mouse. By comparing the ribosome occupancy of mRNAs from these two species within the same cytoplasm, the authors identify transcripts with cis-acting differences between the two species that affect translation. They monitor translation by measuring both the polysomal association of intact mRNAs and by ribosome foot printing, and additionally validate ~50 allele-specific protein level differences by mass spectrometry. Consistent translation measurements using these different approaches - particularly the validation using pulse labeling to quantify protein synthesis - represent a strength of the present study.

One persistent question in studies of translational regulation is the extent to which transcriptional and translational regulatory divergence of the same gene tend to reinforce or compensate. Here, the authors find that, between these two mouse species, compensatory effects dominate. As protein production represents the major phenotypic outcome for gene expression, the buffering of changes in mRNA abundance by alterations in translation suggests that most selection is negative and stabilizing, though in certain specific biological processes there is evidence for positive selection. The authors also report that the number and magnitude of cis-acting changes affecting translation are similar to those affecting transcription, providing a new perspective on an active debate about the relative impact of transcriptional and post-transcriptional control. Finally, the authors find evidence that changes in RNA structure around the start codon, which are implicated in gene-to-gene differences in translation, also contribute to divergence in translation.

This work advances our understanding of the ways that cis-regulatory divergence impacts protein abundance as well as providing some insight into the molecular basis of these translational changes. It is situated within the broader questions regarding the contributions of transcriptional and post-transcriptional control to protein abundance, and it provides these answers in the context of a mammalian systems. I thus expect it will draw broad interest.

1. The authors compare only two species and see only the endpoint of the divergence between them. Despite this, they discuss compensatory regulatory changes in a way that suggests they have data about the order of these events. Notably, in the abstract, they say that their results suggest "a role of the translational regulation in buffering transcriptional noise and thereby maintaining the robustness of protein expression." This statement in fact raises two serious concerns:

A. The authors have no evidence that the translational change post-dates or is driven by the transcriptional change. It could equally be the case that changes in transcriptional regulation buffer the drift in translational control in order to maintain the robustness of protein expression.

B. Noise is different than interspecific (or even intraspecific) genetic divergence. Noise is the stochastic (or in some contexts, environmentally determined) fluctuation in gene expression amongst genetically identical individuals. Certain kinds of gene expression noise, driven by small number fluctuations in mRNA counts, can actually change specifically in response to buffered changes in transcription and translation.

2. The plot in 2C uses mRNA abundance measurements on the denominator of both the x and the y axes. It isn't clear whether exactly the same mRNA-Seq data were used for both quantities. If so, this creates the possibility of spurious correlation, as any variation (noise) in this mRNA-Seq measurement will cause a similar change in both the polysome and the footprinting TE.

3. In 6B, the authors plot mRNA abundance changes against TE changes. This analysis forms the basis for all of Figure 6. However, if the mRNA abundance measurements on the x axis are the same ones that are used as a denominator for the TE measurements on the y axis, then the appearance of compensatory change emerges as an artifact of noise (measurement error) in mRNA abundance measurements. The typical gene shows no allele-specific divergent expression at the level of transcription or translation. However, accurate measurement of identical translation combined with measurement error in mRNA results in an apparent compensatory divergence in transcription and translation.

4. In 3D, it's not clear to me from the description whether the significance calls (green boxes)

adequately reflect for multiple hypothesis testing. Does $p < 0.05$ mean that only 50 / 1000 permutations had a Spearman correlation stronger than the specific value observed in any square on the graph? Or merely in the same square on the graph, in which case we would expect ~5% of squares to be labelled green by chance.

5. In 2D the authors measure few enough allele-specific proteins (just 54) that they could capture the data better by plotting each point rather than box and whiskers summaries.

1st Revision - authors' response

12 June 2015

Reviewer #1:

This manuscript by Hou et al. addresses an important question. The range of data generated and analyzed is impressive, and it is presented in a straightforward way. My only substantive question is, was there any correction for the number of statistical tests performed? For example in Fig 3D, it appears ~250 tests were performed, so ~10-15 would be expected to reach $p < 0.05$ just by chance. Using a more stringent cutoff in this and other figures with multiple tests (e.g. Fig 3C) may help to highlight the truly significant results. By the same token, the result on out of frame uAUGs is based on a p-value of 0.038, which would not be significant after correcting for even just the four uORF/uAUG-related tests in Fig 3E-G. If these results could be replicated with the ribosome profiling data, they would be greatly strengthened; but if the same trend is not seen there, I suspect they may be false positives, in which case the claims about them made in the abstract and elsewhere could be toned down.

A: We would like to thank the referee for appreciating our manuscript and for pointing out the important multiple-testing issue, which we have not carefully addressed in the previous version. In this revision, as suggested by the referee, we performed further analyses, which were detailed below.

In Figure 3C, we compared SNP density enrichment in 5'UTR proximal to the start codon between genes with significant ADTE and control genes without ADTE, and observed that the genes with significant ADTE, compared to the control genes, exhibited on average higher SNP enrichment in the region proximal to the start codon. We used a sliding window to scan the regions proximal to the start codon, and performed then permutation tests for the 20 windows separately. Although five windows have nominal significant P-values (< 0.05), after BH adjustment due to multiple hypothesis testing, no windows remain with a BH-adjusted $P < 0.05$. We consider this might be due to the relatively small size of the ADTE gene group. Therefore we increased the number of genes with ADTE by relaxing the allelic TE divergence fold change cutoff from 2.0 to 1.5, repeating the same analysis again, and found seven windows with BH-adjusted $P < 0.05$, and even three of them with BH-adjusted $P < 0.001$. Now in Figure E4, we show the comparison result (similar as Figure 3A-C) using ADTE genes with the allelic TE divergence fold change cutoff of 1.5. We have edited the text accordingly, and toned down our conclusion in the revised abstract as well as main text.

In Figure 3D, we tested the correlation between ADTE and the allelic difference in the minimum free energy (MEF) of mRNA segments surrounding the start codon. Since the window size and position might confound the result, instead of 'carefully' choosing one window as often done in some other studies, we considered here a whole set of windows with a range of sizes and positions, and in total performed the correlation test for 264 windows. In the previous version of manuscript, we calculated the P value for each window based on the permuted datasets derived from individual corresponding windows. Among the 264 fragments, 35 had a nominal p-value < 0.05 , significantly more than expected by chance (expected ~13; $P = 5.9 \times 10^{-8}$, exact binomial test), indicating that it is very unlikely that all of them were false positives. Now in the revision, as also suggested by the referee #2, to address the multiple-testing issue, we calculated the false discovery rate (FDR) based on the whole permuted datasets derived from all 264 windows. Now, the lowest FDR for observed correlation coefficient was around 25% and therefore we did not claim the statistical significance any more in the revised manuscript.

In Figure 3E-G, we consider these tests were independently performed, therefore correction for multiple-testing does not apply here. We agree with the referee that the effect is marginal and therefore have toned down our conclusion in the revision.

As suggested by the reviewer, we also performed the set of analyses using ribosome footprinting data. As shown in Figure E5, the results based on ribosome footprinting data demonstrated the same trend as observed with polysome profiling data, greatly strengthening our conclusion. We have edited the manuscript accordingly.

Reviewer #2:

In this manuscript, Hou and colleagues analyze the genetic basis of gene expression variability, incorporating translational as well as transcriptional effects, in an interspecies hybrid mouse. By comparing the ribosome occupancy of mRNAs from these two species within the same cytoplasm, the authors identify transcripts with cis-acting differences between the two species that affect translation. They monitor translation by measuring both the polysomal association of intact mRNAs and by ribosome footprinting, and additionally validate ~50 allele-specific protein level differences by mass spectrometry. Consistent translation measurements using these different approaches - particularly the validation using pulse labeling to quantify protein synthesis - represent a strength of the present study.

One persistent question in studies of translational regulation is the extent to which transcriptional and translational regulatory divergence of the same gene tend to reinforce or compensate. Here, the authors find that, between these two mouse species, compensatory effects dominate. As protein production represents the major phenotypic outcome for gene expression, the buffering of changes in mRNA abundance by alterations in translation suggests that most selection is negative and stabilizing, though in certain specific biological processes there is evidence for positive selection. The authors also report that the number and magnitude of cis-acting changes affecting translation are similar to those affecting transcription, providing a new perspective on an active debate about the relative impact of transcriptional and post-transcriptional control. Finally, the authors find evidence that changes in RNA structure around the start codon, which are implicated in gene-to-gene differences in translation, also contribute to divergence in translation.

This work advances our understanding of the ways that cis-regulatory divergence impacts protein abundance as well as providing some insight into the molecular basis of these translational changes. It is situated within the broader questions regarding the contributions of transcriptional and post-transcriptional control to protein abundance, and it provides these answers in the context of a mammalian systems. I thus expect it will draw broad interest.

A: We would like to thank the referee for appreciating the high quality of our data and the importance of our study.

1. The authors compare only two species and see only the endpoint of the divergence between them. Despite this, they discuss compensatory regulatory changes in a way that suggests they have data about the order of these events. Notably, in the abstract, they say that their results suggest "a role of the translational regulation in buffering transcriptional noise and thereby maintaining the robustness of protein expression." This statement in fact raises two serious concerns:

A. The authors have no evidence that the translational change post-dates or is driven by the transcriptional change. It could equally be the case that changes in transcriptional regulation buffer the drift in translational control in order to maintain the robustness of protein expression.

*A: We thank the referee for pointing out this issue. Indeed, we cannot distinguish whether translational regulation compensates transcription, or *vice versa*. Therefore we edited the text in the abstract and main text accordingly.*

B. Noise is different than interspecific (or even intraspecific) genetic divergence. Noise is the stochastic (or in some contexts, environmentally determined) fluctuation in gene expression amongst genetically identical individuals. Certain kinds of gene expression noise, driven by small number fluctuations in mRNA counts, can actually change specifically in response to buffered changes in transcription and translation.

A: We agree that the term ‘noise’ used here can be misleading. Therefore we removed it and changed the text accordingly.

2. The plot in 2C uses mRNA abundance measurements on the denominator of both the x and the y axes. It isn't clear whether exactly the same mRNA-Seq data were used for both quantities. If so, this creates the possibility of spurious correlation, as any variation (noise) in this mRNA-Seq measurement will cause a similar change in both the polysome and the footprinting TE.

A: In Figure 2C, we indeed used the same mRNA-seq data as denominator in both axes. However, the observed correlation is not spurious since the allelic difference in translational status measured by polysome profiling and ribosome footprinting alone also correlated very well ($R^2=0.55$). We now added this plot as Figure E3A.

3. In 6B, the authors plot mRNA abundance changes against TE changes. This analysis forms the basis for all of Figure 6. However, if the mRNA abundance measurements on the x axis are the same ones that are used as a denominator for the TE measurements on the y axis, then the appearance of compensatory change emerges as an artifact of noise (measurement error) in mRNA abundance measurements. The typical gene shows no allele-specific divergent expression at the level of transcription or translation. However, accurate measurement of identical translation combined with measurement error in mRNA results in an apparent compensatory divergence in transcription and translation.

A: To avoid drawing any wrong conclusion due to measurement errors, we have carefully estimated and controlled the experimental noise in our study not only by the bootstrapping test but also through biological replicates. In Figure 4B, the mean values from the two replicates were used. The residuals from the averaged values, which can largely represent the measurement noise, was on average nearly two orders of magnitude lower than the mean of observed allelic divergence in mRNA abundance from the two replicates (the median of residuals: 0.0070; the median of the mean allelic divergence in mRNA abundance: 0.46; $P < 2.2e-16$, Mann-Whitney U test; see also Figure E1C for the correlation between two replicates of allelic divergence in mRNA abundance measurement).

To address the concern of the referee, we assessed the influence by such measurement noise by performing a simulation. First, we shuffled the gene labels for ADTE and allelic mRNA divergence values, therefore removing the biologically meaningful correlation between the two. As shown in Figure R1A, these shuffled data did not show any enrichment of compensatory changes. Then, we started to add "noise" randomly sampled from the residuals from the averaged allelic mRNA divergence; again, no enrichment of compensatory changes was found with this level of "noise" (Figure R1B). To be more conservative, we increased the simulated noise to 10 times of the observed noise level, and still observed no obvious enrichment (Figure R1C). The spurious anti-correlation appeared after further enlarging the noise level to 100 times of the observed residues (Figure R1D). Therefore, we are confident that the observed compensatory allelic divergence between transcription and translation was not caused by measurement errors in mRNA abundance.

Figure R1. Simulation of measurement errors.

4. In 3D, it's not clear to me from the description whether the significance calls (green boxes) adequately reflect for multiple hypothesis testing. Does $p < 0.05$ mean that only 50 / 1000 permutations had a Spearman correlation stronger than the specific value observed in any square on the graph? Or merely in the same square on the graph, in which case we would expect $\sim 5\%$ of squares to be labelled green by chance.

A: In the previous version of manuscript, we calculated the P value for each window based on the permuted datasets derived from individual corresponding windows. Among the 264 fragments, 35 had a nominal p-value < 0.05 , significantly more than expected by chance (expected ~ 13 ; $P=5.9 \times 10^{-8}$, exact binomial test), indicating that it is very unlikely that all of them were false positives. Now in the revision, as suggested by the referee, to address the multiple-testing issue, we calculated FDR based on the whole permuted datasets derived from all 264 windows. The lowest FDR was around 25% and therefore we did not claim the statistical significance any more in the revised manuscript. We have now made the description of our statistical test more clear in the revised Materials and Methods.

5. In 2D the authors measure few enough allele-specific proteins (just 54) that they could capture the data better by plotting each point rather than box and whiskers summaries.

A: Now, we added the scatterplot as Figure E3B. The correlation of the allelic divergence measured by polysome profiling and proteomics is significant ($r=0.82$, $P=5.5 \times 10^{-14}$).

Thank you again for sending us your revised manuscript. We have now heard back from the reviewers and they are now satisfied with the modifications made. I am pleased to inform you that your paper has been accepted for publication.

Reviewer #1:

The authors have satisfied my concern about multiple test correction, and appear to have also addressed the concerns of the other reviewer.

Reviewer #2:

The authors have addressed my concerns with the submitted manuscript. The use of polysome profiling and simulation analysis exclude the possibility that spurious correlations drive the apparent compensatory changes in transcription and translation, which represented the major conceptual concern with the paper. The revised manuscript also corrects more minor concerns with statistical analysis and with interpretation.